# HighRes-net: Multi-Frame Super-Resolution by Recursive Fusion

## Abstract

Generative deep learning has sparked a new wave of Super-Resolution (SR) algorithms that enhance single images with impressive aesthetic results, albeit with imaginary details. Multi-frame Super-Resolution (MFSR) offers a more grounded approach to the ill-posed problem, by conditioning on multiple low-resolution views. This is important for satellite monitoring of human impact on the planet – from deforestation, to human rights violations – that depend on reliable imagery. To this end, we present HighRes-net, the first deep learning approach to MFSR that learns its sub-tasks in an end-to-end fashion: (i) **co-registration**, (ii) **fusion**, (iii) **up-sampling**, and (iv) **registration-at-the-loss**. Co-registration of low-res views is learned implicitly through a reference-frame channel, with no explicit registration mechanism. We learn a global fusion operator that is applied recursively on an arbitrary number of low-res pairs. We introduce a *registered loss*, by learning to align the SR output to a ground-truth through ShiftNet. We show that by learning deep representations of multiple views, we can super-resolve low-resolution signals and enhance Earth observation data at scale. Our approach recently topped the European Space Agency's MFSR competition on real-world satellite imagery.

## 1 Introduction

Multiple low-resolution images collectively contain more information than any individual low-resolution image, due to minor geometric displacements, e.g. shifts, rotations, atmospheric turbulence, and instrument noise. Multi-Frame Super-Resolution (MFSR) (Tsai, 1984) aims to reconstruct hidden high-resolution details from multiple low-resolution views of the same scene. Single Image Super-Resolution (SISR), as a special case of MFSR, has attracted much attention in the computer vision, machine learning and deep learning communities in the last 5 years, with neural networks learning complex image priors to upsample and interpolate images (Xu et al., 2014; Srivastava et al., 2015; He et al., 2016). However, in the meantime not much work has explored the learning of representations for the more general problem of MFSR to address the additional challenges of co-registration and fusion of multiple low-resolution images.

This paper explores how Multi-Frame Super-Resolution (MFSR) can benefit from recent advances in learning representations with neural networks. To the best of our knowledge, this work is the first to introduce a deep-learning approach that solves the co-registration, fusion and registration-at-the-loss problems in an end-to-end learning framework.

Prompting this line of research is the increasing drive towards planetary-scale Earth observation to monitor the environment and human rights violations. Such observation can be used to inform policy, achieve accountability and direct on-the-ground action, e.g. within the framework of the Sustainable Development Goals (Jensen & Campbell, 2019).

**Nomenclature** *Registration* is the problem of estimating the relative geometric differences between two images (e.g. due to shifts, rotations, deformations). *Fusion*, in the MFSR context, is the problem of mapping multiple low-res representations into a single representation. By *co-registration*, we mean the problem of registering all low-resolution views to improve their fusion. By *registration-at-the-loss*, we mean the problem of registering the super-resolved reconstruction

to the high-resolution ground-truth prior to computing the loss. This gives rise to the notion of a *registered loss*.

Co-registration of multiple images is required for longitudinal studies of land change and environmental degradation. The fusion of multiple images is key to exploiting cheap, high-revisit-frequency satellite imagery, but of low-resolution, moving away from the analysis of infrequent and expensive high-resolution images. Finally, beyond fusion itself, super-resolved generation is required throughout the technical stack: both for labeling, but also for human oversight (Drexler, 2019) demanded by legal context (Harris et al., 2018).

**Summary of contributions**

- HighRes-net: We propose a deep architecture that learns to fuse an arbitrary number of low-resolution frames with implicit co-registration through a reference-frame channel.
- ShiftNet: Inspired by HomographyNet (DeTone et al., 2016), we define a model that learns to register and align the super-resolved output of HighRes-net, using ground-truth high-resolution frames as supervision. This registration-at-the-loss mechanism enables more accurate feedback from the loss function into the fusion model, when comparing a super-resolved output to a ground truth high resolution image. Otherwise, a MFSR model would naturally yield blurry outputs to compensate for the lack of registration, to correct for sub-pixel shifts and account for misalignments in the loss.
- By combining the two components above, we contribute the first architecture to learn fusion and registration end-to-end.
- We test and compare our approach to several baselines on real-world imagery from the PROBA-V satellite of ESA. Our performance has topped the Kelvins competition on MFSR, organized by the Advanced Concepts Team of ESA (Märtens et al., 2019) (see section 5).

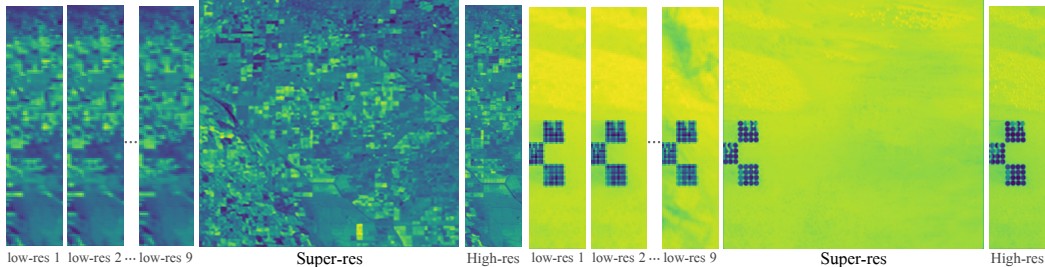

low-res 1 low-res 2 ⋯ low-res 9      Super-res      High-res    low-res 1 low-res 2 ⋯ low-res 9      Super-res      High-res

Figure 1: HighRes-net combines many low-resolution images (300 meters/pixel) into one image of superior resolution. The same site shot in high-resolution (100m/pix) is also shown for reference. Source of low-res and high-res: `imgset1087` and `imgset0285` of PROBA-V dataset, see section 5.

The rest of the paper is divided as follows: in Section 2, we discuss related work on SISR and MFSR; Section 3 outlines HighRes-net and section 4 presents ShiftNet, a differentiable registration component that drives our registered loss mechanism during end-to-end training. We present our results in section 5, and in Section 6 we discuss some opportunities for and limitations and risks of super-resolution.

## 2 BACKGROUND

### 2.1 MULTI-FRAME SUPER-RESOLUTION

How much detail can we resolve in the digital sample of some natural phenomenon? Nyquist (1928) observed that it depends on the instrument's sampling rate and the oscillation frequency of the underlying natural signal. Shannon (1949) built a sampling theory that explained Nyquist's observations when the sampling rate is constant (*uniform* sampling) and determined the conditions of *aliasing* in a sample. Figure 2 illustrates this phenomenon.

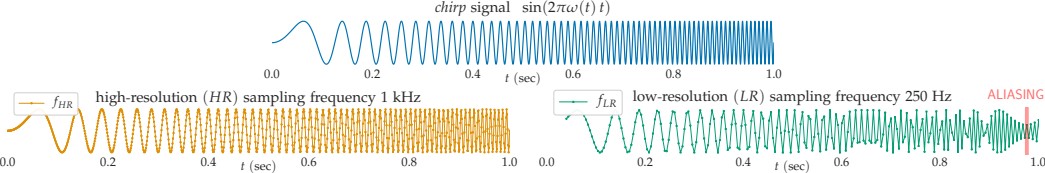

Figure 2: **Top**: A *chirp* harmonic oscillator $\sin(2\pi\omega(t)t)$, with instantaneous frequency $\omega(t)$. **Left**: The shape of the high-resolution sample resembles the underlying chirp signal. **Right**: Close to $t = 1$, the apparent frequency of the low-resolution sample does not match that of the chirp. This is an example of *aliasing* (shown with red at its most extreme), and it happens when the sampling rate falls below the *Nyquist rate*, $s_N = 2 \cdot s_B$, where $s_B$ is the highest non-zero frequency of the signal.

Sampling at high-resolution (left) maintains the frequency of the chirp signal (top). Sampling at a lower resolution (right), this apparent chirped frequency is lost due to aliasing, which means that the lower-resolution sample has a fundamentally smaller capacity for resolving the information of the natural signal, and a higher sampling rate can resolve more information.

Shannon's sampling theory has since been generalized for multiple interleaved sampling frames (Papoulis, 1977; Marks, 2012). One result of the generalized sampling theory is that we can go beyond the Nyquist limit of any individual uniform sample by interleaving several uniform samples taken concurrently. When an image is down-sampled to a lower resolution, its high-frequency details are lost permanently and cannot be recovered from any image in isolation. However, by combining multiple low-resolution images, it becomes possible to recover the original scene at a higher resolution.

Moreover, different low-resolution samples may be sampled at different phases, such that the same high-resolution frequency information will be packed with a phase shift. As a consequence, when multiple low-resolution samples are available, the fundamental challenge of MFSR is de-aliasing, i.e. disentangling the high-frequency components (Tsai, 1984).

The first work on MSFR (Tsai, 1984) considered the reconstruction of a high-resolution image as a fusion of co-registered low-resolution images in the Fourier domain. With proper *registration* and fusion (Irani & Peleg, 1991; Fitzpatrick et al., 2000; Capel & Zisserman, 2001), a composite *super-resolved* image can reveal some of the original high-frequency detail that would not have been accessible from single low-resolution image. In this work, we introduce HighRes-Net, which aims to provide an end-to-end deep learning framework for MFSR settings.

**Relation to Video and Stereo Super-Resolution** While there are obvious similarities to Video SR (Tao et al., 2017; Sajjadi et al., 2018; Yan et al., 2019; Wang et al., 2019b) and Stereo SR (Wang et al., 2019d) (Wang et al., 2019a), the setting of this work differs in several ways: HighRes-net learns to super-resolve sets and not sequences of low-res views. Video SR relies on motion estimation from a sequence of observations. Also, prediction at time $t = T$ relies on predictions at $t < T$ (autoregressive approach). Whereas in our case, we predict a single image from an unordered set of low-res inputs. Also, the low-res views are multi-temporal (taken at different times).

Video SR methods assume that the input is a temporal sequence of frames. Motion or optical flow can be estimated to super-resolve the sequences of frames. In this work, we do not assume low-res inputs to be ordered in time. Our training input is a set of low-res views with unknown timestamps and our target output is a single image — not another sequence.

## 2.2 A PROBABILISTIC APPROACH

In addition to aliasing, MFSR deals with random processes like noise, blur, geometric distortions – all contributing to random low-resolution images. Traditionally, MFSR methods assume a-priori knowledge of the data generating motion model, blur kernel, noise level and degradation process; see for example, Pickup et al. (2006). Given multiple low-resolution images, the challenge of MFSR is to reconstruct a plausible image of higher-resolution that could have generated the observed low-resolution images. Optimization methods aim to improve an initial guess by minimizing an error between simulated and observed low-resolution images. These methods traditionally model the

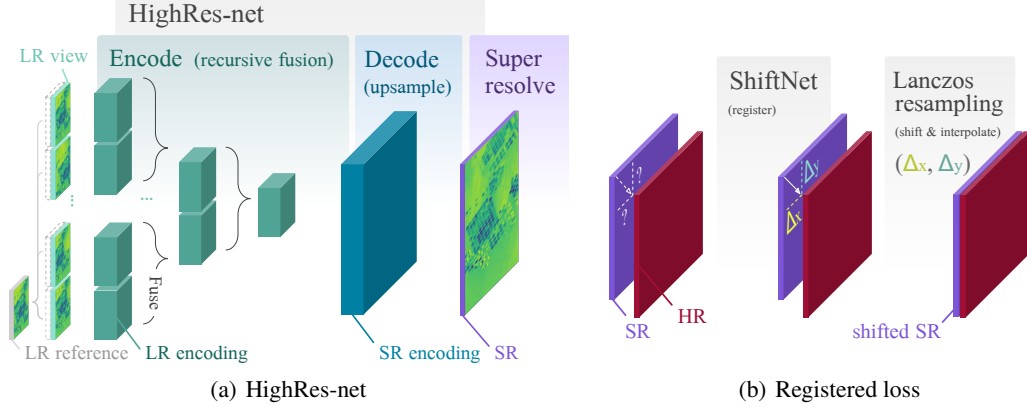

(a) HighRes-net          (b) Registered loss

Figure 3: Schematic of the full processing pipeline, trained end-to-end. At test time, only HighRes-net is used. (a) **HighRes-net**: In the *Encode* stage, an arbitrary number of LR views are paired with the *reference* low-res image (the median low-res in this work). Each LR view–reference pair is encoded into a *view-specific* latent representation. The LR encodings are fused recursively into a single *global* encoding. In the *Decode* stage, the global representation is upsampled by a certain zoom factor ($\times 3$ in this work). Finally, the *super-resolved* image is reconstructed by combining all channels of the upsampled global encoding. (b) **Registered loss**: Generally, the reconstructed SR will be shifted with respect to the ground-truth HR. ShiftNet learns to estimate the $(\Delta x, \Delta y)$ shift that improves the loss. Lanczos resampling: $(\Delta x, \Delta y)$ define two 1D shifting Lanczos kernels that translate the SR by a separable convolution.

additive noise $\epsilon$ and prior knowledge about natural images explicitly, to constrain the parameter search space and derive objective functions, using e.g. Total Variation (Chan & Wong, 1998; Farsiu et al., 2004), Tikhonov regularization (Nguyen et al., 2001) or Huber potential (Pickup et al., 2006) to define appropriate constraints on images.

In some situations, the image degradation process is complex or not available, motivating the development of nonparametric strategies. Patch-based methods learn to form high-resolution images directly from low-resolution patches, e.g. with k-nearest neighbor search (Freeman et al., 2002; Chang et al., 2004), sparse coding and sparse dictionary methods (Yang et al., 2010; Zeyde et al., 2010; Kim & Kwon, 2010)). The latter represents images in an over-complete basis and allows for sharing a prior across multiple sites.

In this work, we are particularly interested in super-resolving satellite imagery. Much of the recent work in Super-Resolution has focused on SISR for natural images. For instance, Dong et al. (2014) showed that training a CNN for super-resolution is equivalent to sparse coding and dictionary based approaches. Kim et al. (2016) proposed an approach to SISR using recursion to increase the receptive field of a model while maintaining capacity by sharing weights. Many more networks and learning strategies have recently been introduced for SISR and image deblurring. Benchmarks for SISR (Timofte et al., 2018), differ mainly in their upscaling method, network design, learning strategies, etc. We refer the reader to (Wang et al., 2019d) for a more comprehensive review.

Few deep-learning approaches have considered the more general MFSR setting and attempted to address it in an end-to-end learning framework. Reecently, Kawulok et al. (2019) proposed a *shift-and-add* method and suggested "including image registration" in the learning process as future work.

In the following sections, we describe our approach to solving both aspects of the registration problem – co-registration and registration-at-the-loss – in a memory-efficient manner.

## 3 HIGHRES-NET: MFSR BY RECURSIVE FUSION

In this section, we present HighRes-net, a neural network for multi-frame super-resolution inside a single spectral band (greyscale images), using joint co-registration and fusion of multiple low-resolution views in an end-to-end learning framework. From a high-level, HighRes-net consists of an encoder-decoder architecture and can be trained by stochastic gradient descent using high-resolution ground truth as supervision, as shown in Figure 3.

**Notation** We denote by $\theta$ the parameters of HighRes-net trained for a given upscaling factor $\gamma$. $LR_{v,i} \in \mathbb{R}^{C \times W \times H}$ is one of a set of $K$ low-resolution views from the same site $v$, where $C$, $W$ and $H$ are the number of input channels, width and height of $LR_{v,i}$, respectively. We denote by $SR_v^\theta = F_\theta^\gamma (LR_{v,1}, \ldots, LR_{v,K})$, the output of HighRes-net and by $HR_v \in \mathbb{R}^{C \times \gamma W \times \gamma H}$ a ground truth high-resolution image. We denote by $[T_1, T_2]$ the concatenation of two images channel-wise. In the following we supress the index $v$ over sites for clarity.

HighRes-Net consists of three main steps: (1) encoding, which learns relevant features associated with each low-resolution view, (2) fusion, which merges relevant information from views within the same scene, and (3) decoding, which proposes a high-resolution reconstruction from the fused summary.

## 3.1 ENCODE, FUSE, DECODE

**Embed, Encode** The core assumption of MFSR is that the low-resolution image set contains collectively more information than any single low-resolution image alone, due to differences in photometric or spatial coverage for instance. However, the redundant low frequency information in multiple views can hinder the training and test performance of a MFSR model. We thus compute a reference image *ref* as a *shared representation* for multiple low-resolution views $(LR_i)_{i=1}^K$ and embed each image jointly with *ref*. This highlights differences across the multiple views (Sanchez et al., 2019), and potentially allows HighRes-net to focus on difficult high-frequency features such as crop boundaries and rivers during super-resolution. The *shared* representation or *reference* image intuitively serves as an anchor for implicitly aligning and denoising multiple views in deeper layers. We refer to this mechanism as *implicit co-registration*.

HighRes-net's embedding layer emb$_\theta$ consists of a convolutional layer and two residual blocks with PReLu activations (He et al., 2015) and is shared across all views. The embedded hidden states $s_i^0$ are computed in parallel as follows:

$$ref\,(c, i, j) = \text{median}\,(LR_1(c, i, j), \ldots, LR_K(c, i, j)), \quad \text{such that } ref \in \mathbb{R}^{C \times W \times H} \quad (1)$$
$$s_i^0 = \text{emb}_\theta\,([LR_i,\ ref]) \in \mathbb{R}^{C_h \times W \times H}, \quad (2)$$

where $C_h$ denotes the channels of the hidden state.

The imageset is padded if the number of low-res views $K'$ is not a power of 2: we pad the set with dummy zero-valued views, such that the new size of the imageset $K$ is the next power of 2. See Algorithm 1, line 1.

**Fuse** The embedded hidden states $s_i^0$ are then fused recursively, halving by two the number of low-resolution states at each fusion step $t$, as shown in Figure 4. Given a pair of hidden states $s_i^t, s_j^t$, HighRes-net computes a new representation:

$$\left[\tilde{s}_i^t,\ \tilde{s}_j^t\right] = \left[s_i^t,\ s_j^t\right] + g_\theta\left(\left[s_i^t,\ s_j^t\right]\right) \in \mathbb{R}^{2C_h \times W \times H} \quad (3)$$
$$s_i^{t+1} = s_i^t + \alpha_j f_\theta\left(\tilde{s}_i^t,\ \tilde{s}_j^t\right) \in \mathbb{R}^{C_h \times W \times H}, \quad (4)$$

where $\tilde{s}_i^t,\ \tilde{s}_j^t$ are intermediate representations; $g_\theta$ is a shared-representation within an inner residual block (equation 3); $f_\theta$ is a fusion block, and $\alpha_j$ is 0 if the $j$-th low-resolution view is part of the padding, and 1 otherwise. $f_\theta$ squashes $2C_h$ input channels into $C_h$ channels and consists of a (conv2d+PReLu). Intuitively, $g_\theta$ *aligns* the two representations and it consists of two (conv2d + PReLU) layers.

The blocks $(f_\theta, g_\theta)$ are shared across all pairs and depths, giving it the flexibility to deal with variable size inputs and significantly reduce the number of parameters to learn.

**Upscale and Decode** After $T = \log_2 K$ fusion layers, the final low-resolution encoded state $s_i^T$ contains information from all $K$ input views. Any information of a spatial location that was initially missing from $LR_i$, is now encoded implicitly in $s_i^T$. $T$ is called the depth of HighRes-net. Only then, $s_i^T$ is upsampled with a deconvolutional layer (Xu et al., 2014) to a higher-resolution space $s_{HR}^T \in \mathbb{R}^{C_h \times \gamma W \times \gamma H}$. The hidden high-resolution encoded state $s_{HR}^T$ is eventually convolved with a $1 \times 1$ 2D kernel to produce a final super-resolved image $SR^\theta \in \mathbb{R}^{C \times \gamma W \times \gamma H}$.

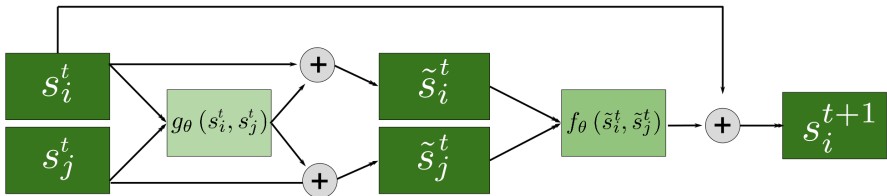

Figure 4: HighRes-net's global fusion operator consists of a *co-registration* $g_\theta$ and a *fusion* $f_\theta$ block which aligns and combines two representations into a single representation.

The overall architecture of HighRes-net is summarized in Figure 3(a) and the pseudocode for the forward pass is given in Algorithm 1.

---

**Algorithm 1:** HighRes-net forward pass

---

**Input:** low-res views $LR_1 \ldots LR_{K'}$

1   $(LR_1 \ldots LR_K, \alpha_1 \ldots \alpha_K) \leftarrow \text{pad}(LR_1 \ldots LR_{K'})$      // pad inputs to next power of 2

2   $s_i^0 \leftarrow \text{encode}(LR_i)$      // parallelized across $1 \ldots K$

3   $T \leftarrow \log_2 K$      // fusion depth

4   $k \leftarrow K$

5   **for** $t = 1 \ldots T$ **do**

6      **for** $i = 1 \ldots k/2$ **do**

7        $s_i^t \leftarrow \text{fuse}(s_i^{t-1}, s_{k-i}^{t-1}, \alpha_{k-i})$      // fuse encoded views

8      $k = k/2$

9   $SR \leftarrow \text{decode}(s_i^T)$

**Output:** super-resolved view $SR$

---

## 4   REGISTRATION MATTERS

Co-registration matters for fusion. HighRes-net learns to implicity co-register multiple low-resolution views $LR_i$ and fuse them into a single super resolved image $SR_\theta$.

A more explicit registration-at-the-loss can also be used for measuring similarity metrics and distances between $SR_\theta$ and $HR$. Indeed, training HighRes-Net alone, by minimizing a reconstruction error such as the mean-squared error between $SR_\theta$ and $HR$, leads to blurry outputs, since the neural network has to compensate for pixel and sub-pixel misalignments between its output $SR_\theta$ and $HR$.

Here, we present ShiftNet-Lanczos, a neural network that can be paired with HighRes-net to account for pixel and sub-pixel shifts in the loss, as depicted in Figure 3(b). Our ablation study A.2 and qualitative visual analysis suggest that this strategy helps HighRes-net learn to super-resolve and leads to clearly improved results.

### 4.1   SHIFTNET-LANCZOS

ShiftNet learns to align a pair of images with sub-pixel translations. ShiftNet registers pairs of images by predicting two parameters defining a global translation. Once a sub-pixel translation is found for a given pair of images, it is applied through a Lanczos shift kernel to align the images.

**ShiftNet** The architecture of ShiftNet is adapted from HomographyNet (DeTone et al., 2016). Translations are a special case of homographies. In this sense, ShiftNet is simply a special case of HomographyNet, predicting 2 shift parameters instead of 8 homography parameters. See Appendix A.3, for details on the architecture of ShiftNet.

---

**Algorithm 2:** Sub-pixel registered loss through ShiftNet-Lanczos

---

**Input:** $SR_\theta$, $HR$       // super-resolved view, high-resolution ground-truth

1   $(\Delta x, \Delta y) \leftarrow \text{ShiftNet}(SR_\theta, HR)$       // register $SR$ to $HR$

2   $\kappa_\Delta \leftarrow \text{LanczosShiftKernel}(\Delta)$       // 1D Lanczos kernels for $x$ and $y$ sub-pixel shifts

3   $SR_{\theta,\Delta} \leftarrow SR_\theta * \kappa_{\Delta x} * \kappa_{\Delta y}$       // 2D sub-pixel shift by separable 1D convolutions

4   $\ell_{\theta,\Delta} \leftarrow \text{loss}(SR_{\theta,\Delta}, HR)$       // sub-pixel registered loss

**Output:** $\ell_{\theta,\Delta}$

---

One major difference from HomographyNet is the way we train ShiftNet: In (DeTone et al., 2016), HomographyNet is trained on synthetically transformed data, supervised with ground-truth homography matrices. In our setting, ShiftNet is trained to cooperate with HighRes-net, towards the common goal of MFSR (see section **Objective function** below).

**Lanczos shift & interpolation kernel**    To shift and align an image by a sub-pixel amount, it must be convolved with a filter that shifts for the integer parts and interpolates for the fractional parts of the translation. Standard options for interpolation include the nearest-neighbor, sinc, bilinear, bicubic, and Lanczos filters (Turkowski, 1990). The sinc filter has an infinite support as opposed to any digital signal, so in practice it produces ringing or ripple artifacts — an example of the Gibbs phenomenon. The nearest-neighbor and bilinear filters do not induce ringing, but strongly attenuate the higher-frequency components (over-smoothing), and can even alias the image. The Lanczos filter reduces the ringing significantly by using only a finite part of the since (up to a few lobes from the origin). Experimentally, we found the Lanczos filter to perform the best.

**Objective function**    In our setting, registration benefits super-resolution. HighRes-net receives more informative gradient signals when its output is aligned with the ground truth high-resolution image. Conversely, super-resolution benefits registration, since good features are key to align images (Clement et al., 2018). We thus trained HighRes-Net and ShiftNet-Lanczos in a cooperative setting, where both neural networks work together to minimize an objective function, as opposed to an adversarial setting where a generator tries to fool a discriminator. HighRes-net infers a latent super-resolved variable and ShiftNet maximises its similarity to a ground truth high-resolution image with sub-pixel shifts.

By predicting and applying sub-pixel translations in a differentiable way, our approach for registration and super-resolution can be combined in an end-to-end learning framework. Shift-Net predicts a sub-pixel shift $\Delta$ from a pair of high-resolution images. The predicted transformation is applied with Lanczos interpolation to align the two images at a pixel level. ShiftNet and HighRes-Net are trained jointly to minimize a common loss function, using backpropagation and stochastic gradient descent. Our objective function is composed of a registered reconstruction loss computed as in Algorithm 2. In our case, we used the corrected clear PSNR metric (cPSNR), chosen by ESA, which is a variant of the mean squared error, designed to correct for brightness and clouds in satellite images (Märtens et al., 2019), but the proposed architecture is decoupled from the choice of loss.

See Algorithm 4 for the pseudo-code for computing the alignment loss $\ell_{\theta,\Delta}$. We further regularize the L2 norm of ShiftNet's ouput with a hyperparameter $\lambda$ and our final joint objective is given by:

$$L_{\theta,\Delta}(SR_\theta, HR) = \ell_{\theta,\Delta} + \lambda ||\Delta||_2. \tag{5}$$

## 5   EXPERIMENTS AND RESULTS

Prior SR work has focused on super-resolving low-res images that are artificially generated by simple bilinear down-sampling, (Bulat et al., 2018) The PROBA-V satellite has separate cameras onboard for capturing high-res / low-res pairs. As far as we know, the PROBA-V dataset is the first publicly available dataset for MFSR that contains naturally occurring low-res and high-res pairs. This is in contrast to most of the work in SR (SISR, MFSR, Video SR, Stereo SR) that synthetically down-sample high-res images and frames (Wang et al., 2019c; Nah et al., 2019). Methods that are

trained on artificially downscaled datasets fail to produce good results when applied to real-world low-resolution, low quality images (Shocher et al., 2018). For this reason, we experiment only on PROBA-V, a dataset that does not suffer from biases induced by artificial down-sampling.

## 5.1 PROBA-V KELVIN DATASET

The performance of our method is illustrated with satellite imagery from the Kelvin competition, organized by ESA's Advanced Concept Team (ACT).

The Proba-V Kelvin dataset (Märtens et al., 2019) contains 1450 scenes (RED and NIR spectral bands) from 74 hand-selected Earth regions around the globe at different points in time. The scenes are split into 1160 scenes for training and 290 scenes for testing. Each data-point consists of exactly one 100m resolution image as $384 \times 384$ grey-scale pixel images (HR) and several 300m resolution images from the same scene as $128 \times 128$ grey-scale pixel images (LR), spaced days apart. We refer the reader to the Proba-V manual (Wolters et al., 2014) for further details on image acquisition.

Each scene comes with at least 9 low-res views, and an average of 19. Each view comes with a noisy *quality map*. The quality map is a binary map, that indicates concealed pixels due to volatile features, such as clouds, cloud shadows, ice, water and snow. The sum of clear pixels (1s in the binary mask) is defined as the *clearance* of a low-res view. These incidental and noisy features can change fundamental aspects of the image, such as the contrast, brightness, illumination and landscape features. We use the *clearance* scores to randomly sample from the imageset of low-res views, such that views with higher *clearance* are more likely to be selected. This strategy helps to prevent overfitting. See Appendix A.4 for more details.

**Working with missing or noisy values** A quality map can be used as a *binary mask* to indicate noisy or occluded pixels, due to clouds, snow, or other volatile objects. Such a mask can be fed as an additional input channel in the respective low-res view, in the same fashion as thereference frame. When missing value masks are available, neural networks can learn which parts of the input are anomalous, noisy, or missing, when provided with such binary masks (see e.g. Che et al. (2018)). In satellite applications where clouds masks are not available, other segmentation methods would be in order to infer such masks as a preprocessing step (e.g. Long et al. (2015)). In the case of the PROBA-V dataset, we get improved results when we make no use of the masks provided. Instead we use the masks only to inform the sampling scheme within the low-res imageset to prevent overfitting.

## 5.2 EXPERIMENTS

Across all experiments, we used the same hyperparameters, reported in Appendix A.1. By default, each imageset is padded to 32 views for training and testing, unless specified otherwise. Our pytorch implementation requires less than 9h of training on a single NVIDIA V100 GPU. At test time, super-resolving an imageset of size 128x128 by a factor of 3, takes less than 0.2 seconds. Our code is made available on github[1].

We evaluated different models on ESA's Kelvin competition. Our best model, HighRes-Net trained jointly with shiftNet-Lanczos, scored consistently at the top of the public and final leaderboard, see Table 1. In the following, we discuss several baselines and report our experiments.

## 5.3 COMPARISONS

- ESA baseline  upsamples each low-resolution view separately with bicubic up-sampling and averages those of maximum clearance.
- SRResNet  (Ledig et al., 2017) is a deep learning SISR baseline.
- SRResNet-1 + shiftNet  was trained with ShiftNet.
- SRResNet-6 + shiftNet  differs from the previous model during test time only. It independently upsamples 6 low-resolution views with *SRResNet-1*, co-registers the super-resolved images using *shiftNet*, and averages the 6 aligned super-resolved images into a final prediction.
- ACT baseline   (Märtens et al., 2019) is a Convolutional Neural Network with five fixed channels for the five clearest low-resolution views.

---

[1] https://anonymous.4open.science/r/b3404d0d-e541-4f52-bbe9-f84f2a52972e/

- DeepSUM baseline (Molini et al., 2019) can be seen as a variant of *SRResNet-6 + shiftNet*. Multiple low-res views are independently upsampled, then co-registered and *fused* into a single image.
- HighRes-net + shiftNet are described in sections 3 and 4. Upsampling is done in the last step as opposed to (Molini et al., 2019).
- Ensemble   An ensemble of two trained (HighRes-net + shiftNet) models, one with K=16 and one with K=32 input views, whose outputs are averaged.

## 5.4 ESA KELVIN LEADERBOARD

The Kelvin competition used the *corrected clear* PSNR (cPSNR) quality metric as the standardized measure of performance. The cPSNR is a variant of the Peak Signal to Noise Ratio (PSNR) used to compensate for pixel-shifts and brightness bias. We refer the reader to (Märtens et al., 2019) for the motivation and derivation of this quality metric. The cPSNR metric is normalized by the score of the ESA baseline algorithm so that a score smaller than 1 means "better than the baseline" and lower is better. We also use it as our training objective with sub-pixel registration (see also section 3(b) on ShiftNet).

Table 1: Public and final leaderboard scores for the ESA's Kelvin competition. Lower score means a better reconstruction.

| Method | Public score | Final score |
|---|---|---|
| SRResNet (Ledig et al., 2017) | 1.0095 | 1.0084 |
| ESA baseline | 1.0000 | 1.0000 |
| SRResNet-1 + shiftNet | 1.0002 | 0.9995 |
| ACT baseline (Märtens et al., 2019) | 0.9874 | 0.9879 |
| SRResNet-6 + shiftNet | 0.9808 | 0.9794 |
| HighRes-net + shiftNet **(ours)** | 0.9496 | 0.9488 |
| HighRes-net + shiftNet Ensemble **(ours)** | **0.94738** | 0.94774 |
| DeepSUM (Molini et al., 2019) | 0.94884 | **0.94744** |

### 5.4.1 ABLATION STUDY

We further ran an ablation study on the available labeled data (1450 image sets), split in 90% / 10% for training and testing. Our results suggest that more low-resolution views benefit the reconstruction error, plateuing after 16 views, see Appendix A.2. Another finding is that registration matters for MFSR, both in co-registering low-res views, and registering-at-the-loss, see Appendix A.3. Finally, selecting the $k$ clearest views for fusion can lead to ovefitting. One remedy is to randomly sample the views with a bias for clearance, see A.4.

## 6 DISCUSSION

### 6.1 THE IMPORTANCE OF GROUNDED DETAILS

The PROBA-V satellite (Dierckx et al., 2014) was launched by ESA to monitor Earth's vegetation growth, water resources and agriculture. As a form of data fusion and enrichment, multi-frame super-resolution could enhance the vision of such satellites for scientific and monitoring applications (Carlson & Ripley, 1997; Pettorelli et al., 2005). More broadly, satellite imagery can help NGOs and non-profits monitor the environment and human rights (Cornebise et al., 2018; Helber et al., 2018; Rudner et al., 2019; Rolnick et al., 2019) at scale, from space, ultimately contributing to the UN sustainable development goals. Low-resolution imagery is cheap or sometimes free, and it is frequently updated. However, with the addition of fake or imaginary details, such enhancement wouldn't be valuable as scientific, legal, or forensic evidence.

### 6.2 FUTURE WORK

Registration matters at the loss stage but also at the fusion stage. The latter is not explicit in our model and the reason why and how it works is less understood. Learning to sample a reference frame

and learning to fuse multiple representations with attention could also be a promising approach to extend HighRes-net.

Ensuring authenticity of detail is a major challenge and quantifying uncertainty of super-resolved images is an important line of future work for real world applications. Along this line of research, the question of how to evaluate a super-resolved image is important for downstream tasks and, more generally, similarity metrics remain an open question for many computer visions tasks (Bruna et al., 2015; Johnson et al., 2016; Isola et al., 2017; Ledig et al., 2017).

## 6.3 CONCLUSION

In this paper, we presented HighRes-net – the first deep learning approach to multi-frame super-resolution that learns typical sub-tasks of MFSR in an end-to-end fashion: (i) **co-registration**, (ii) **fusion**, (iii) **up-sampling**, and (iv) **registration-at-the-loss**.

It recursively fuses a variable number of low-resolution views by learning a global fusion operator. The overall fusion also aligns all low-resolution views with an implicit co-registration mechanism through the reference channel. We also introduced ShiftNet-Lanczos, a network that learns to register and align the super-resolved output of HighRes-net with a high-resolution ground-truth.

Registration is important both to align multiple low-resolution inputs (co-registration) and to compute similarity metrics between shifted signals. Our experiments suggest that an end-to-end cooperative setting (HighRes-net + ShiftNet-Lanczos) helps with training and test performance. By design, our approach is fast to train, faster to test, and low in terms of memory-footprint by doing the bulk of the computational work (co-registration + fusion) on multiple images while maintaining their low-resolution height & width.

There is an ongoing proliferation of low-resolution yet high-revisit low-cost satellite imagery, but they often lack the detailed information of expensive high-resolution imagery. We believe MFSR can raise its potential to NGOs and non-profits that contribute to the UN Sunstainable Development Goals.

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

# A   APPENDIX

## A.1   EXPERIMENTAL DETAILS

We trained our models on low-resolution patches of size 64x64. HighRes-net's architecture is reported in Table 3. We denote by Conv2d(in, out, k, s, p) a conv2D layer with *in* and *out* input/output channels, kernel of size (k,k), stride (s,s) and padding p. We used the ADAM optimizer (Kingma & Ba, 2014) with default hyperparameters and trained our models on batches of size 32, for 400 epochs, using 90% of the data for training and 10% for validation. Our learning rate is initialized to 0.0007, decayed by a factor of 0.97 if the validation loss plateaus for more than 2 epochs. For the regularization of shiftNet, we employed $\lambda = 0.000001$.

Table 2: ResidualBlock(h) architecture

| layer0 | Conv2d(in=h, out=h, k3, s1, p1) |
|--------|----------------------------------|
| layer1 | PReLU |
| layer2 | Conv2d(in=h, out=h, k3, s1, p1) |
| layer3 | PReLU |

Table 3: HRNet architecture

| Step | Layers | Number of params |
|------|--------|------------------|
| encode | Conv2d(in=2, out=64, k3, s1, p1) | 1216 |
| | PReLU | 1 |
| | ResidualBlock(64) | 73858 |
| | ResidualBlock(64) | 73858 |
| | Conv2d(in=64, out=64, k3, s1, p1) | 36928 |
| fuse | ResidualBlock(128) | 295170 |
| | Conv2d(in=128, out=64, k3, s1, p1) | 73792 |
| | PReLU | 1 |
| decode | ConvTranspose2d(in=64, out=64, k3, s1) | 36928 |
| | PreLU | 1 |
| | Conv2d(in=64, out=1, k1, s1) | 65 |
| residual (optional) | Upsample(scale_factor=3.0, mode=bicubic) | 0 |
| | | **591818** (total) |

Thanks to weight sharing, HighRes-net super-resolves scenes with 32 views in 5 recursive steps, while requiring less than 600K parameters. ShiftNet has more than 34M parameters (34187648) but is dropped during test time. We report GPU memory requirements in table 4 for reproducibility purposes.

Table 4: GPU memory requirements to train HighRes-net + ShiftNet on patches of size 64x64 with batches of size 32, and a variable number of low-resolution frames.

| # views | 32 | 16 | 4 |
|---------|----|----|---|
| GPU memory (GB) | 27 | 15 | 6 |

## A.2   HOW MANY FRAMES DO YOU NEED?

We trained and tested HighRes-net with ShiftNet using 1 to 32 frames. With a single image, our approach performs worse than the ESA baseline. Doubling the number of frames significantly improves both our training and validation scores. After 16 frames, our model's performance stops increasing as show in Figure 5.

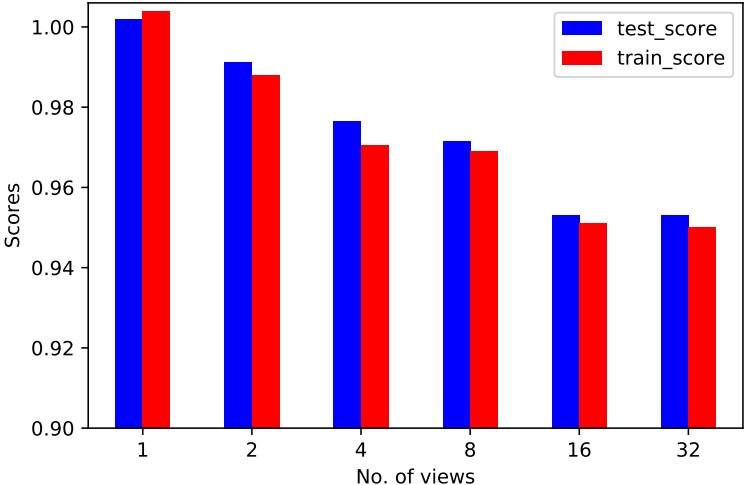

Figure 5: Public leaderboard scores vs. nviews for HighRes-net + ShiftNet. Lower is better.

## A.3 REGISTRATION MATTERS

**Registered loss** The only explicit registration that we perform is at the loss stage, to allow the model partial credit for a solution. This solution can be enhanced but otherwise mis-registered wrt to the ground truth. We trained our base model HighRes-net without ShiftNet-Lanczos and observed a drop in performance as shown in Table 5. Registration matters and aligning outputs with targets helps HighRes-net generate sharper outputs and achieve competitive results.

Table 5: Registration matters: Train and validation scores for HighResNet trained with and without ShiftNet-Lanczos. Lower is better.

| HighRes-net + | train score | test score |
|---|---|---|
| w/o registration | 0.9616 | 0.9671 |
| ShiftNet + Lanczos | **0.9501** | **0.9532** |

**Implicit co-registration** The traditional practice in MFSR is to explicitly co-register the LR views prior to super-resolution (Tsai, 1984; Molini et al., 2019). The knowledge of sub-pixel miss-alignments tells an algorithm what pieces of information to fuse from each LR image for any pixel in the SR output. Contrary to the conventional practice in MFSR, we propose implicit co-registeration by pairing LR views with a reference frame a.k.a. anchor. In this sense, we never explicitly compute the relative shifts between any LR pair. Instead, we simply stack each view with a chosen reference frame as an additional channel to the input. We call this strategy *implicit co-registration*, We found this strategy to be effective in the following ablation study which addresses the impact of the choice of a reference frame aka *anchor*.

We observe the median reference is the most effective in terms of train and test score. We suspect the median performs better than the mean because the median is more robust to outliers and can help denoise the LR views. Interestingly, training and testing without a shared reference performed worse than the ESA baseline. This shows that co-registration (implicit or explicit) matters. This can be due to the fact that the model lacks information to align and fuse the multiple views.

**ShiftNet architecture** ShitNet has 8 layers of (`conv2D` + `BatchNorm2d` + `ReLU`). Layer 2, 4 and 6 are followed by `MaxPool2d`. The final output is flattened to a vector $x$ of size 32768. Then, we compute a vector of size 1024, $x = \text{ReLU}(\text{fc1}(\text{dropout}(x)))$. The final shift prediction is

Table 6: Train and validation scores for HighRes-net + ShiftNet-Lanczos trained and tested with different references as input. Lower is better.

| Reference | train score | test score |
|---|---|---|
| None (no co-registration) | 1.0131 | 1.0088 |
| Mean of 9 LRs | 0.9636 | 0.9690 |
| Median or 9 LRs (base) | **0.9501** | **0.9532** |

$\texttt{fc2}(x)$ of size 2. The bulk of the parameters come from $\texttt{fc1}$, with $32768 \times 1024$ weights. These alone, account for 99% of ShiftNet's parameters. Adding a $\texttt{MaxPool2d}$ on top of layer 3, 5, 7 or 8 halves the parameters of ShiftNet.

### A.4 TOWARDS PERMUTATION INVARIANCE

A desirable property of a fusion model acting on an un-ordered set of images, is permutation-invariance: the output of the model should be invariant to the order in which the LR views are fused. An easy approach to encourage permutation invariant neural networks is to randomly shuffle the inputs at training time before feeding them to a model (Vinyals et al., 2015).

In addition to randomization, we still want to give more importance to clear LR views (with high clearance score), which can be done by sorting them by clearance. A good trade-off between uniform sampling and deterministic sorting by *clearance*, is to sample $k$ LR views without replacement and with a bias towards higher clearance:

$$p(i \mid C_1, \ldots, C_k) = \frac{e^{\beta C_i}}{\sum_{j=1}^{k} e^{\beta C_j}}, \qquad (6)$$

where $k$ is the total number of LR views, $C_i$ is the clearance score of $\text{LR}_i$ and $\beta$ regulates the bias towards higher clearance scores,

When $\beta = 0$, this sampling strategy corresponds to uniform sampling and when $\beta = +inf$, this corresponds to picking the k-clearest views in a deterministic way. Our default model was trained with $\beta = 50$ and our experiments are reported in Table 7.

Table 7: Validation scores vs. nviews for HighRes-net + ShiftNet. Lower is better.

| Sampling strategy | train score | test score |
|---|---|---|
| $\beta = \infty$ (k-clearest) | **0.9386** | 0.9687 |
| $\beta = 0$ (uniform-k) | 0.9638 | 0.9675 |
| $\beta = 50$ (base) | 0.9501 | **0.9532** |

From Table 7, $\beta = \infty$ reaches best training score and worst testing score. For $\beta = 50$ and $\beta = 0$, the train/test gap is much more reduced. This suggests that the deterministic strategy is overfitting and randomness prevents overfitting (diversity matters). On the other hand, $\beta = 50$ performs significantly better than $\beta = 0$ suggesting that biasing a model towards higher clearances could be beneficial i.e., clouds matter too.

