# OpenReview forum: "HighRes-net: Multi-Frame Super-Resolution by Recursive Fusion"
_ICLR.cc/2020/Conference — Reject_

### Official Review · AnonReviewer1 · 2019-10-10
**Official Blind Review #1**

**Rating:** 8

**Review:**

This paper proposes an end-to-end multi-frame super-resolution algorithm, that relies on a pair-wise co-registrations and fusing blocks (convolutional residual blocks), embedded in a encoder-decoder network 'HighRes-net' that estimates the super-resolution image. Because the ground truth SR image is typically misaligned with the estimation SR image, the authors proposed to learn the shift with a neural network 'ShiftNet' in a cooperative setting with HighRes-net. The experiments were performed on the ESA challenge on satellite images, showing good results.

Overall, I found this paper interesting, and the method described is both clever and efficient. While some points need to be clarified, I am in favor of accepting this paper to ICLR.

Positive aspects:
- the paper is very clear and easy to read, with nice figures.
- a sensitivity analysis on many different parameters or types of inputs are made, which makes this paper an interesting research paper. For example, the tests on the type of reference image to stack at the beginning are very interesting.
- While I am not an expert on super-resolution, I do see a clever algorithm, that can be for example used with different number of input views.
- The end-to-end framework is also quite interesting as it allows to be spread easily across the very large satellite images users, with the code aldready publicly available.
- Lastly, the results are good wrt to the state-of-the-art, as the algorithm was proposed during a 2019 challenge and was in the top ones on the private leaderboard.

Remarks and clarifications:
- After looking at the challenge website, it is quite strange to find that the results values in Table 1, in terms of public and private scores, differ from the actual leaderboards, even if the metric is still the same. Thus, I was not able to understand (i) if the authors really participate to the challenge (ii) what method they presented during the challenge as 3 or 4 are shown here (iii) what was their ranking in the real leaderboard. It is important to be precise. From what I see, they might be second on the private score (the one that matters), if so it can be clarified also in the abstract where the word 'topped' was used.
- cPSNR metric : (i) can you please explain the acronym; (ii) it was the competition metric, so it means it is not 'we use' but 'the challenge used' - which will also show that it was not a choice to satisfy your results, but a standard metric.
- Lanczos interpolation: can you please explain in a small sentence in what this interpolation differs from the others?
- ShiftNet: what is this network? We only know tha it is adapted from HomographyNet, but we don't have any information about how it is composed. We just know from App. 1 that it has a very large number of parameters (34M). Why is that? Why a rigid registration needs such a large number of parameters?
- median anchor image: this is one of the interesting points of the paper. Can you please just clarify that you take a naive median image? Such as: for pixel (i,j), med(i,j)= med( LR1(i,j), LR2(i,j) ,...) .
- You are saying 'each imageset is padded to 32 views'. What do you mean? I thought your network was able to be used with different numbers of views. How did you pad your imageset?
- you list 'several baselines': for me, a baseline is something to compare to - usually, it is even something easy, such as the ESA baseline. But in your list, you are mixing baselines, other state-of-the-art methods (some of which you don't beat, so it's not a 'baseline'...), and your methods. It is very difficult to know which ones are yours. Please make different lists and/or identify yours in the table 1.
- What is the Ensemble method? It appears that in the paper, you explain the 'HighRes-net + shiftNet' method, but actually, the 'Ensemble' is the better one, while not clearly described. I don't understand what outputs are averaged - I thought HighRes-net and shiftNet were performed simultaneously.
- How did you select the hyperparameters of your model?


Scientific questions:
- Is it possible to have views of different sizes as inputs? Or views with missing parts?
- Usually, we have a high resolution image that is different in terms of type of acquisition (like a different type of satellite, but also true in medical images for ex.). Is your network ready for that? I mean: can we constrain the method even if it is not the same type of acquisition as ground truth? In that case, is that possible to have a super-resolution of the type of the input LR images?

Typos:
- is comprised of -> is composed of
- in Table 7, the bold number should be the beta=infinity as it is the best one. It will be clearer, even if of course, a good train score does not mean a good method because of overfitting.


**Experience Assessment:**

I have read many papers in this area.

**Review Assessment: Checking Correctness Of Derivations And Theory:**

I assessed the sensibility of the derivations and theory.

**Review Assessment: Checking Correctness Of Experiments:**

I carefully checked the experiments.

**Review Assessment: Thoroughness In Paper Reading:**

I read the paper thoroughly.

---

> ### Author Response · Authors · 2019-11-14
> **Clarifications on the comparisons, leaderboard**
>
> Thank you for your detailed assessment of our work
>
> 1. > “it is quite strange to find that the results values in Table 1, in terms of public and private scores, differ from the actual leaderboards, even if the metric is still the same. Thus, I was not able to understand (i) if the authors really participate to the challenge (ii) what method they presented during the challenge as 3 or 4 are shown here (iii) what was their ranking in the real leaderboard. It is important to be precise. From what I see, they might be second on the private score (the one that matters), if so it can be clarified in the abstract where the word 'topped' was used.”
>
> To clear any confusion, we did participate in the ESA challenge as team Rifat.
> By “topped” we mean “achieved competitive results”. Our ensemble model, an ensemble of two HighRes-net models trained separately (discussed in a comment below) achieved
> - 0.947388637793901 (1st position) on the public leaderboard: https://kelvins.esa.int/proba-v-super-resolution/leaderboard/
> - 0.9477450367529225 (2nd position) on the final leaderboard (public + private): https://kelvins.esa.int/proba-v-super-resolution/results/
>
> The source of the confusion was the rounding up of the scores to 4 decimals - reporting 0.9474 on the Public leaderboard and 0.9477 on the Public+Private.
> DeepSUM achieved the best score (0.9474466476281652) in the final leaderboard.
> To get a sense of the scale, note that the 3rd place scored 0.9576339586408439.
>
> Beyond the cPSNR evaluation metric, our model requires an order of magnitude less training time - 10 hours for HighRes-net VS 1 week for DeepSUM.
> This is because the DeepSUM architecture first learns to upscale each low-res view with a shared SISR-type (Single-Image SR) net.
> Then all the downstream tasks must to be learned on the size of a high-res image - x3 upscaling factor for PROBA-V, which means a x9 increase in memory demand.
>
> To summarize, HighRes-net is competitive in terms of cPSNR, and also an order of magnitude faster to train.
>
> Thank you for pointing out these sources of confusion. We’ll write the scores in their full precision in the revised manuscript.
>
> -----------------
>
> 7. > “you list 'several baselines': for me, a baseline is something to compare to - usually, it is even something easy, such as the ESA baseline. But in your list, you are mixing baselines, other state-of-the-art methods (some of which you don't beat, so it's not a 'baseline'...), and your methods. It is very difficult to know which ones are yours. Please make different lists and/or identify yours in the table 1.”
>
> Thank you for this suggestion. We have identified which lines in Table 1 correspond to our methods.
>
> ------------------
>
> 8. > “What is the Ensemble method? It appears that in the paper, you explain the 'HighRes-net + shiftNet' method, but actually, the 'Ensemble' is the better one, while not clearly described. I don't understand what outputs are averaged - I thought HighRes-net and shiftNet were performed simultaneously.
>
> We can see how this is confusing. Our paper is about the HighRes-net + ShiftNet method indeed.
> For our Ensemble method, we trained two HighRes-net+shiftNet models, one with K=16 and one with K=32 frames.
> We averaged HighRes-Net_16 + HighRes-Net_32 outputs for our predictions on the final leaderboard.
> We have included these details in our revised manuscript.

---

> ### Author Response · Authors · 2019-11-14
> **cPSNR metric**
>
> 2. > “cPSNR metric : (i) can you please explain the acronym; (ii) it was the competition metric, so it means it is not 'we use' but 'the challenge used' - which will also show that it was not a choice to satisfy your results, but a standard metric.”
>
> Indeed, cPSNR (clear corrected Peak to Signal Noise Ratio) is a standardized score used by the challenge as the competition metric.
>
> Also see, https://kelvins.esa.int/proba-v-super-resolution/scoring/ for the justification of the score.
> We also used it as our training objective with sub-pixel registration (see section on ShiftNet).
>
> Thank you for raising this. We'd hate to give the impression that we custom-designed this metric to suit our performance. We have amended our paper accordingly.

---

> ### Author Response · Authors · 2019-11-14
> **Lanczos filter**
>
> 3. > “Lanczos interpolation: can you please explain in a small sentence in what this interpolation differs from the others?”
>
> To shift an image by a fractional amount, it must convolved with a filter that shifts and “interpolates”. Standard candidates filters in Image Processing are the sinc, bi-linear, bi-cubic, and Lanczos. See also. "Filters for common resampling tasks." Turkowski, Graphics gems, 1990.
>
> The sinc filter has an infinite support, so it produces ringing / ripple artifacts (aka Gibbs phenomenon) in practice. The bilinear filter over-smooths / over-attenuates high frequencies - the opposite of ringing but can alias results. The Lanczos filter - with finite support - approximates the sinc filter, and reduces the ringing artifacts. The bi-cubic filter performs similarly to Lanczos in practice but is more difficult to code from scratch.
>
> Thank you for raising this. We have clarified the Lanczos interpolation filter in Section 4.1.

---

> ### Author Response · Authors · 2019-11-14
> **ShiftNet**
>
> 4. > “ShiftNet: what is this network? We only know that it is adapted from HomographyNet, but we don't have any information about how it is composed. We just know from App. 1 that it has a very large number of parameters (34M). Why is that? Why a rigid registration needs such a large number of parameters?”
>
> ShitNet has an architecture of 8 layers made of conv2D + BatchNorm2d + ReLU. Layer 2, 4 and 6 are followed by MaxPool2d. The final output is flattened to a vector x of size 32768. Then, we compute a vector of size 1024: x = ReLU( fc1( dropout( x ) ) ). The final shift prediction is fc2(x) of size 2.
>
> The bulk of the parameters come from fc1, with 32768 * 1024 weights. These alone, account for 99% of ShiftNet's parameters. Adding a MaxPool2d on layer 3, 5, 7 or 8, halves the parameters of ShiftNet roughly by two.
> Our code for ShiftNet was adapted from: https://github.com/mazenmel/Deep-homography-estimation-Pytorch/blob/master/HomographyNet.py
>
> Thank you for raising this question.
> We have clarified the difference between ShiftNet and HomographyNet in Section 4.1: ShiftNet. We have also included details on the architecture of ShiftNet, in a new paragraph in Appendix A.3.
>
> Please also see our comment to R2: https://openreview.net/forum?id=HJxJ2h4tPr&noteId=BylRdPcOjB

---

> ### Author Response · Authors · 2019-11-14
> **Median reference frame**
>
> 5. > “median anchor image: this is one of the interesting points of the paper. Can you please just clarify that you take a naive median image? Such as: for pixel (i,j), med(i,j)= med( LR1(i,j), LR2(i,j) ,...).”
>
> Yes, we simply take a median image the way you suggested. This is not median filtering as pointed by Reviewer3.
>
> We have clarified this in equation 1, section 3.1.

---

> ### Author Response · Authors · 2019-11-14
> **Padding with zero-valued images**
>
> 6. > “You are saying 'each imageset is padded to 32 views'. What do you mean? I thought your network was able to be used with different numbers of views. How did you pad your imageset?”
>
> - There is a typo in Algorithm 1, line 1: the right-hand-side should be K’, not K
> (LR1 ... LRK, α1 ... αK) = pad (LR1 ... LRK’), where the original input size K’ can be any number of low-res views.
>
> Consider an example with K = 16 low-res views. Once encoded, the 16 encoding tensors are grouped into pairs, and each pair is fused into one encoding. The pairing and fusing repeats until all pairs are fused into a single universal encoding. In this case, K is a power of 2 so no padding is needed.
>
> If K’ is not a power of 2, then we pad the set of low-res views with dummy zero-valued views, such that the new size K is a power of 2.
>
> The fact that we end up padding each imageset to 32 views is coincidental. An imageset in the PROBA-V dataset can contain up to 32 low-res views. So for PROVA-V, we typically have to pad the imageset up to 32 frames. Hence, even though HighRes-net can handle a variable number of low-res views, in practice for PROBA-V all imagesets end up with the same number of views prior to the forward pass. A power of 2 makes the recursive fusion more efficient, hence why we experimented with an upper limit of 1, 2, 4, 8, 16, 32 views in our ablation study (Appendix A.2).
>
> Thank you for raising this ambiguity. To clarify this, we added a paragraph in Section 3.1, after equation 2.

---

> ### Author Response · Authors · 2019-11-14
> **Hyperparameters**
>
> 9. > How did you select the hyperparameters of your model?
>
> For the network design, we used design choices common in the SR literature that we have cited.
> For upsampling, we used a stride = kernel = 3, which made sense for the upsampling factor of 3 needed for the competition, otherwise checkerboard artifacts become more frequent.
> The learning rate and its decay schedule, were chosen by trial and error.

---

> ### Author Response · Authors · 2019-11-14
> **Scientific questions**
>
> 10. > “Is it possible to have views of different sizes as inputs? Or views with missing parts?”
>
> This question is orthogonal to a lot of the deep learning computer vision literature.
> Our model is trained on patches of size 64x64 and tested on images of size 128x128.
> A simplistic way to handle multiple inputs of different size is by padding the image dimensions to max(height, width), and indicate the padded parts with a binary (missing value) mask.
> Reframed as a missing-value problem, then our model can handle views with missing parts (e.g. concealed pixels because of clouds or padding) with a missing-value channel.
> Please see also Part 6 of our comment to R3: https://openreview.net/forum?id=HJxJ2h4tPr&noteId=ryxbd5DuiH
>
> ----------------
>
> 11. > “Usually, we have a high resolution image that is different in terms of type of acquisition (like a different type of satellite, but also true in medical images for ex.). Is your network ready for that? I mean: can we constrain the method even if it is not the same type of acquisition as ground truth? In that case, is it possible to have a super-resolution of the type of the input LR images?”
>
> Yes we can. High-res and low-res images were obtained from different cameras onboard the PROBA-V satellite.
> Please, see also Part 4 of our comment to R3:
> https://openreview.net/forum?id=HJxJ2h4tPr&noteId=ryxbd5DuiH
>
> If “different” here means in spectral band, whether that is between input and output, or within inputs, then the general problem of fusion of hyperspectral / multispectral images for down-stream prediction tasks - but not MFSR - has been studied extensively in remote sensing.
> For a starting point, see e.g. Helber, et al. (2018) in our references.
>
> As far as we know, MFSR with inputs from different band is an open problem.
>
> ----------------
>
> "Typos"
>
> Typos have been fixed.
>
> Thank you for your detailed analysis of our paper. We have added all clarifications in our revised manuscript.

---

### Official Review · AnonReviewer2 · 2019-10-23
**Official Blind Review #2**

**Rating:** 3

**Review:**

The paper proposes a framework including recursive fusion to co-registration and registration loss to solve the problem that the super-resolution results and the high-resolution labels are not pixel aligned.  Besides, the method is able to achieve good performance in the Proba-V Kelvin dataset. However, I have some concerns about this paper:

1) This paper lacks many references. Recently, many works focus on multi-frame super-resolution containing video super-resolution and stereo image super-resolution via deep learning.  They are using multiple low-resolution image to construct high-resolution image. For example:

Stereo super-resolution:

Jeon, Daniel S., et al. "Enhancing the spatial resolution of stereo images using a parallax prior." *Proceedings of the IEEE Conference on Computer Vision and Pattern Recognition*. 2018.

Wang, Longguang, et al. "Learning parallax attention for stereo image super-resolution." *Proceedings of the IEEE Conference on Computer Vision and Pattern Recognition*. 2019.

Video super-resolution:

Tao, Xin, et al. "Detail-revealing deep video super-resolution." *Proceedings of the IEEE International Conference on Computer Vision*. 2017.

FRVSR: Sajjadi, Mehdi SM, Raviteja Vemulapalli, and Matthew Brown. "Frame-recurrent video super-resolution." *Proceedings of the IEEE Conference on Computer Vision and Pattern Recognition*. 2018.

FFCVSR: Yan, Bo, Chuming Lin, and Weimin Tan. "Frame and Feature-Context Video Super-Resolution." *Proceedings of the AAAI Conference on Artificial Intelligence*. Vol. 33. 2019.

EDVR: Wang, Xintao, et al. "Edvr: Video restoration with enhanced deformable convolutional networks." *Proceedings of the IEEE Conference on Computer Vision and Pattern Recognition Workshops*. 2019.

2) Recursive fusion is aimed to fuse multiple low-resolution image information. Recently, more and more work utilize different methods to fuse multiple low-resolution image. For example, Tao et al proposes SPMC (Sub-pixel Motion Compensation) to align image, FRVSR uses unsupervised flow network that predicts optical flow to warp image, FFCVSR directly concatenate low-resolution image as the input of 2D convolutional network to fuse the information, and EDVR fuses multiple image features via utilizing deformable convolution. Thus, what is the advantage of recursive fusion compared to the above methods? This paper should discuss the difference between recursive fusion and the above methods.

3) Registration loss is important in this paper and it can solve the problem the output SR is not pixel-wise aligned to the HR ground truth. Registration loss utilizes ShiftNet that is adapted from HomographyNet. Thus, what is the difference between ShiftNet and HomographyNet? This paper should add some details about ShiftNet and Lanczos interpolation.

4) It is better to test more datasets and compare with more state-of-the-art methods. This paper only tests in a satellite image dataset. Some datasets can be considered such as VID4 dataset in video super-resolution.

**Experience Assessment:**

I have published in this field for several years.

**Review Assessment: Checking Correctness Of Derivations And Theory:**

N/A

**Review Assessment: Checking Correctness Of Experiments:**

I carefully checked the experiments.

**Review Assessment: Thoroughness In Paper Reading:**

I read the paper thoroughly.

---

> ### Author Response · Authors · 2019-11-12
> **Thank you for reviewing our work**
>
> 1. > “This paper lacks many references. Recently, many works focus on MFSR containing video SR and stereo image SR via deep learning.”
>
> Thank you for pointing out these references. We've included them in our revision.
> We want to stress that our setting is different from video SR in several ways:
>
> - We learn to super-resolve sets and not sequences of low-res views. Video SR relies on motion estimation from a sequence of observations. Also, prediction at time t=T relies on predictions at t<T (autoregressive approach). Whereas in our case, we predict a single image from an unordered set of low-res inputs.
>
> - In our setting, the low-res views are multi-temporal (taken at different times) from different revisits. Please see Paragraph 1 in our comment to Reviewer 3: https://openreview.net/forum?id=HJxJ2h4tPr&noteId=B1l6sKmUoB
>
> Our work different from Stereo SR:
> - Multi-temporality (see above)
> - These images were taken at a nadir direction (top-down view) above different subpoints (coordinates below satellite). Whereas Stereo SR assumes that both views focus on the same point, but from different angles.
>
> ---
>
> 2. > “Thus, what is the advantage of recursive fusion compared to the above methods [Video SR papers: SPMC, FRVSR, FFCVSR, EDVR]? This paper should discuss the difference between recursive fusion and the above methods.”
>
> SPMC, FRVSR, FFCVSR and EDVR are all video SR algorithms. They all assume the input to be a temporal sequence of frames. Motion or optical flow can be estimated to super-resolve the sequences of frames. (see Part 1 of this comment above). In this work, we do not assume low-res inputs to be ordered in time. Our training input is a set of low-res views with unknown timestamps and our target output is a single image - not another sequence. More importantly, those Video SR methods were trained and tested on synthetically down-scaled data - we elaborate on this on Part 4 below.
>
> DeepSUM (Molini et al.) is another method that scored similarly to ours in the same ESA competition leaderboard with the same dataset. Their paper already showed competitive results compared to an architecture inspired by DUF ("Deep video SR network using dynamic upsampling filters without explicit motion compensation." CVPR, 2018)
>
> Thank for raising this important question. We've included this in the Related Work discussion.
>
> ---
>
> 3. > “It is better to test more datasets and compare with more state-of-the-art methods. This paper only tests in a satellite image dataset. Some datasets can be considered such as VID4 dataset in video super-resolution.”
>
> Many benchmarks exist for Super Resolution:
> VIDEO SR:
> - Vid4: C. Liu and D. Sun. A bayesian approach to adaptive video super resolution. In CVPR, pages 209–216. IEEE, 2011.
> - Vimeo 90K: Xue, Tianfan, et al. "Video enhancement with task-oriented flow." International Journal of Computer Vision 127.8 (2019): 1106-1125.
> - Y10: Sajjadi, Mehdi SM, Raviteja Vemulapalli, and Matthew Brown. "Frame-recurrent video super-resolution." Proceedings of the IEEE Conference on Computer Vision and Pattern Recognition. 2018.
> - REDS4 (NTIRE 2019): Nah, Seungjun, et al. "Ntire 2019 challenge on video deblurring: Methods and results." Proceedings of the IEEE Conference on Computer Vision and Pattern Recognition Workshops. 2019.
>
> SISR:
> - Set 5: M. Bevilacqua, A. Roumy, C. Guillemot, and M. L. AlberiMorel. Low-complexity single-image super-resolution based on nonnegative neighbor embedding. 2012.
> - Set 14: R. Zeyde, M. Elad, and M. Protter. On single image scale-up using sparse-representations. In International Conference on Curves and Surfaces, pages 711–730. Springer, 2010.
>
> STEREO:
> - FLICKR 1024: Wang, Yingqian, et al. "Flickr1024: A Large-Scale Dataset for Stereo Image Super-Resolution." CVPR Workshops 2019
>
> However, in all of these datasets, the low-res views are artificially down-scaled, and prior work has shown such methods do not generalize to real world low-res imagery.
> See also paragraph 4 of our comment to Reviewer3: https://openreview.net/forum?id=HJxJ2h4tPr&noteId=ryxbd5DuiH
>
> “The vast majority of prior work for this problem focus on how to increase the resolution of low-resolution images which are artificially generated by simple bilinear down-sampling (or in a few cases by blurring followed by down-sampling). We show that such methods fail to produce good results when applied to real-world low-resolution, low quality images” - https://arxiv.org/abs/1807.11458.
> See also, Zero shot paper. Shocher, Assaf, Nadav Cohen, and Michal Irani. "“zero-shot” super-resolution using deep internal learning." CVPR 2018.
>
> We agree that benchmarking on more datasets and with more models would strengthen our paper. We are not aware of other publicly available datasets for MFSR with real low-res (not artificially down-sampled) images. By working on this dataset, we want to encourage the community to consider on real-world (not synthetic) images for super-resolution.

---

> ### Author Response · Authors · 2019-11-12
> **On ShiftNet and Lanczos**
>
> > “What is the difference between ShiftNet and HomographyNet? This paper should add some details about ShiftNet and Lanczos interpolation.”
>
> The architecture of ShiftNet is not novel and not the major contribution of this paper. ShiftNet and HomographyNet differ in the number of outputs predicted (2 shift parameters in our case vs. 8 homography parameters in the original paper). Translations are a special case of homographies. In this sense, ShiftNet is simply a special case of HomographyNet.
>
> More importantly, ShiftNet differs from HomographyNet in that it is trained in a cooperative setting with HighResNet for MFSR (see section 4.1, Objective Function). The original HomographyNet paper trained in a supervised setting, on synthetically transformed data, using ground truth homography matrices for supervision.
>
> Thank you for raising the need to clarify ShiftNet and Lanczos interpolation.
>
> We have clarified the difference between ShiftNet and HomographyNet and Lanczos interpolation in Section 4.1.
> We have also included details on the architecture of ShiftNet, in a new paragraph in Appendix A.3.
>
>
> Please also see our comment to Reviewer1 on ShiftNet:
> https://openreview.net/forum?id=HJxJ2h4tPr&noteId=BkgWk9QjiH
> and our comment on Lanczos:
> https://openreview.net/forum?id=HJxJ2h4tPr&noteId=SylUYtXjiH

---

### Official Review · AnonReviewer3 · 2019-10-24
**Official Blind Review #3**

**Rating:** 1

**Review:**

This paper presents a multi-frame super-resolution method applied to satellite imagery. It first estimates a reference image for the multiple input LR images by median filtering. Then it pairwise encodes the reference image and each of the multiple images in a recursive fashion then fuses the corresponding feature maps with residual blocks and bottleneck layers until only one feature maps for the entire multiple images obtained. In other words, LR images are fused into a single global encoding. Then, it applies a standard upsampling network to obtain the super-resolved image this image is fed into a network that estimates only the translational shift, and the shifted image with the estimated translation parameters finally resampled.

A major concern is the estimation of a single translational motion for the SR image at the end of the network after all multiple images are already fused. The fusion strategy disregards the underlying spatially varying motion. This explicitly assumes the images are on a flat surface, which perhaps an acceptable assumption for high-orbit satellite imagery where the ground surface depth variances might be negligible. Still, this is a very critical limitation of the method. Besides, I am not convinced that pair-wise fusion can handle significant translational fusion as the filters have shared parameters. How a single convolutional layer accomplishes a global encoding and compensates for any translation between any LR image pair is neither articulated nor convincing discussion and evaluations are provided. Of course, such a problematic approach needs at least some kind of motion compensation, which may explain the need for the ShiftNet layer at the end. Nevertheless, this seems quite problematic.

Even assuming the method only applies to satellite imagery, it lacks mechanisms to compensate/distinguish cloud coverage and atmospheric distortions. Characterization of satellite imagery noise models (Weibull, etc.) common in such imagery as a prior also completely disregarded. For these reasons, the proposed method fails to be considered as a comprehensive approach for multi-image super-resolution of satellite imagery.

Novelty-wise, there is very little as all modules have been commonly used for SR tasks.

**Experience Assessment:**

I have published in this field for several years.

**Review Assessment: Checking Correctness Of Derivations And Theory:**

I carefully checked the derivations and theory.

**Review Assessment: Checking Correctness Of Experiments:**

I carefully checked the experiments.

**Review Assessment: Thoroughness In Paper Reading:**

I read the paper thoroughly.

---

> ### Author Response · Authors · 2019-11-07
> **Request for clarification**
>
> Thank you for reviewing our work.
>
> Can you please clarify what you mean by the following sentence?
> > "The fusion strategy disregards the underlying spatially varying motion"
>
> Are you suggesting that the fusion strategy disregards the relative translation between the low-res views?

---

> > ### Comment · AnonReviewer3 · 2019-11-07
> > **Absolutely, I would love to provide more explanation about my comment**
> >
> > As far as I am aware, there is no low orbit geostationary (fixed) satellite for imaging. The multiple low-resolution images used in the paper are obtained from the PROBA-V, which has 101 minutes orbit period. This is a (relatively fast) moving satellite. In other words, the low-resolution images are acquired at different geocentric coordinates. This means there is a baseline between any two low-resolution images (images are taken at different positions). Now, let's think about the translational motion. A translational only motion as used in the paper can model if only the motion of the camera is exactly parallel to the imaged planar surface. When the surface is not planar, the observed motion at each pixel would have a different motion even if the only observed motion is due to the satellite motion. For instance, a pixel corresponding to a mountain peak and another one corresponding to a valley floor will have very different translational motion coefficients (delta_x, telta_y) even every other thing on the surface is static. The fact is that the earth's surface is not flat. It is not a planar surface. The apparent pixel-wise motion due to the different topology of the earth's surface is what I referred to as the "spatially varying motion". The proposed method only estimates two translational coefficients per low-resolution image pair, thus completely disregards the effects of the non-flat earth surface, which changes the relative motion at every pixel depending on their distance from the camera imaging plane. Also, the proposed method disregards the other common factors that cause pixel-wise observable motion between two images such as moving vehicles, waves, cloud movements, cast shadow changes due to earth rotation, atmospheric turbulence artifacts (heat-induced image warping), etc. I hope this provides some clarification. Thanks.

---

> > > ### Author Response · Authors · 2019-11-11
> > > **Thank you for your detailed and prompt response**
> > >
> > > We address some of your comments below:
> > >
> > > 1. “This is a (relatively fast) moving satellite. In other words, the low-resolution images are acquired at different geocentric coordinates.”
> > >
> > > The orbital velocity is not the main source of variation of geolocation during the low-res acquisition.
> > >
> > > Satellites typically acquire only one low-res image per revisit of some coordinate (principal point). PROBA-V captures images at the nadir direction (top-down view) above a given coordinate (subpoint = principal point). So for a given coordinate (site) PROBA-V acquires one low-res view per 101 mins. The low-res views of a site are not taken in burst mode during a single flyover. In fact, in the ESA dataset the LR views are spaced days apart, see section 5.1.
> > >
> > > Any translational component in the motion between low-res views is due to noisy geolocation: satellites rely on noisy radio signals from ground stations to know their position, and therefore for knowing when to capture an image.
> > >
> > > --------
> > >
> > > 2. "The apparent pixel-wise motion due to the different topology of the earth's surface is what I referred to as the "spatially varying motion"."
> > >
> > > Thank you for your clarification on "spatially varying motions". The technical term for the effect you are describing is "parallax" - a core concept in Remote Sensing, Photogrammetry and Stereo SR, see for instance:
> > > - Learning Parallax Attention in Stereo Super-Resolution, CVPR 2019
> > > - Enhancing the Spatial Resolution of Stereo Images using a Parallax Prior, CVPR 2018
> > >
> > > You are correct in that if 32 low-res views were acquired during a single fly-over, the successive geolocations would indeed be significantly different and the parallax effect would be magnified. This is the case in aerial photography, for instance.
> > >
> > > --------
> > >
> > > 3. “The proposed method only estimates two translational coefficients per low-resolution image pair”
> > >
> > > This is a misunderstanding: Our proposed method estimates two translational coefficients only for the super-resolved image (the final predicted output). Low-res co-registration is handled by the reference frame, together with our fusion block, which consists of three convolution layers (and not a single layer as reported in your original review).
> > >
> > > --------
> > >
> > > 4. “thus completely disregards the effects of the non-flat earth surface, which changes the relative motion at every pixel depending on their distance from the camera imaging plane”
> > >
> > > Please note that the recursive fusion stage accepts only the encoded low-res / reference pairs. So nothing in our co-registration scheme explicitly assumes that the difference in low-res images must be explained by translational motion only. The only stage where translation parameters are explicitly required, is to translate the SR output so as to accurately measure the image similarity in our loss function.  The final translational registration in the end does not directly affect the output, but indirectly by "registering" the loss towards a less biased estimate. Without registration-at-the-loss, the model would compensate for all possible motion between the SR prediction and the ground truth, which would result in a blurry output, see Table 5, Appendix A.3 for the ablation experiment.
> > >
> > > As a crude analogy, we like to think of the registered-loss as a way to endow the model with corrective spectacles. This way, the error gradients carry a more meaningful feedback from the loss. Once fully trained (at test time), our method does not rely on ShiftNet or rigid translations.
> > >
> > > --------
> > >
> > > 5. "Also, the proposed method disregards the other common factors that cause pixel-wise observable motion between two images such as moving vehicles, waves, cloud movements, cast shadow changes due to earth rotation, atmospheric turbulence artifacts (heat-induced image warping), etc."
> > >
> > > As we explain in paragraph 1 of this comment, the low-res images are spaced days apart. So short-lived motions like those of vehicles, waves, or clouds, would require require a video-like time-lapse of images across the time-scale of seconds, minutes, hours. This type of problem requires video super-resolution algorithms. These assume the input to be a sequence of frames, ordered by time. This way, motion or optical flow can be estimated and used to super resolve the sequences of frames. In our framework, we assume low-res observations as an unordered set.
> > >
> > > Other sources of motion, such as cast shadows, and atmospheric turbulence are difficult to model but we do not disregard them. Instead of hard-coding them as rules or domain-expert knowledge, we let our model learn to fuse images by implicit co-registration which is not limited to rigid transformation like pure translations (as we explain in paragraph 4 of this comment).
> > >
> > > We hope that this clarifies our problem setting and the scope of our solution.

---

> ### Author Response · Authors · 2019-11-08
> **Can you please elaborate on your novelty assessment?**
>
> Our work would benefit from an expanded discussion on its novelty. Can you please elaborate on this statement?
>
> > "Novelty-wise, there is very little as all modules have been commonly used for SR tasks."
>
> Novelty-wise, the use of neural networks as modules for SISR and MFSR tasks is common indeed these days (convolution, deconvolution, transpose convolution...).
>
> However, our work differs from SISR by conditioning the output on multiple inputs. Our work differs from video super resolution, in that we do not assume the frames or samples to be in a particular order. To our best knowledge, our fusion block is novel, offering an approach to combine information from multiple frames without requiring temporal order or motion models. Our full architecture could handle an arbitrarily large and unordered set of low-res inputs (unlike DeepSum e.g.).
>
> Here is an exhaustive list of novelties in our work:
> (a) implicit co-registration via a reference frame
> (b) recursive fusion
> (c) a differentiable registered loss in a deep MFSR setting
> (d) a deep MSFR architecture that accepts an arbitrarily large and unordered set (not just a sequence) of low-res views
> (e) sampling on the set of low-res views to prevent over-fitting
> (f) ablation experiments that demonstrate the efficacy of the above
> (g) through a public competition have we demonstrated SOTA performance on the new Proba-V dataset from the European Space Agency, for Earth Observation and Vegetation Growth monitoring - a unique dataset for MFSR in remote sensing.
>
> 1. Besides the upsampling module, can you please specify the modules you referred to as non-novel, and which papers have used them for SR tasks?
>
> 2. If any, can you please specify related methods we should therefore compare to?
>
> 3. The biggest novelty is arguably the consolidation of all the above elements to achieve a deep MFSR in an end-to-end fully differentiable fashion. As far as we know, no other work has done this.

---

> ### Author Response · Authors · 2019-11-12
> **Discussion of concerns on parallax**
>
> Thank you for pointing out the limitations of this paper. Below we address these concerns and provide some clarifications:
>
> 1. > "The apparent pixel-wise motion due to the different topology of the earth's surface is what I referred to as the "spatially varying motion"." (from comment https://openreview.net/forum?id=HJxJ2h4tPr&noteId=BJgRT5MMsr )
>
> Thank you for your clarification on "spatially varying motions". The technical term for the effect you are describing is "parallax" - a core concept in Remote Sensing, Photogrammetry and Stereo SR, see for instance the papers that Reviewer2 cited:
> - Learning Parallax Attention in Stereo Super-Resolution, CVPR 2019
> - Enhancing the Spatial Resolution of Stereo Images using a Parallax Prior, CVPR 2018
>
> You are correct in that if 32 low-res views were acquired during a single fly-over, the successive geolocations would indeed be significantly different and the parallax effect would be magnified. This is the case in aerial photography, for instance.
>
> ---
>
> 2. > "This explicitly assumes the images are on a flat surface, which perhaps an acceptable assumption for high-orbit satellite imagery where the ground surface depth variances might be negligible. Still, this is a very critical limitation of the method."
>
> The parallax p is inversely proportional to the distance d from the object (see e.g. Zeilik and Gregory, 1998):
> p ∝ 1 / d.
>
> If 32 low-res views were acquired during a single fly-over, the successive geolocations would indeed be significantly different and the parallax effect would be magnified. This is the case in aerial photography, for instance.
>
> In our case, we are interested in detecting vegetation growth (PROBA-V), road networks (infrastructure), farms / ranches (agriculture), deforestation (Amazon) or human presence and buildings. In all these monitoring applications, the objects of interests are no more than, say, 50 m tall (for trees).
>
> The parallax effect between low-res images does not inhibit the super-resolution of the previous objects. The lowest of LEO altitudes is 300 km (PROBA-V is about 800km), so the relative depth variation is at most 50m / 300,000m = 0.0033%. The parallax effect is imperceptible for 50m tall objects. Below are some calculations to support this claim:
> Given a point A at height 50m (distance d_A = 300,000 - 50), and a point B at height 0 (distance d_B = 300,000m), their relative change in motion is: p_A / p_B = d_B / d_A = 300,000 / 299,950 = 30 / 29.995.
> This means that if point A moved 30m, then point B moved 5 millimeters less than 30m due to parallax.
> In the case of a fast LEO satellite like PROBA-V, its geolocation is accurate enough such that the translational shifts are mostly within a sub-pixel accuracy, and they almost never exceed 2 pixels. On the ground, 2 pixels amount to a baseline length of (2 px) * (300 m/px) = 600m.
> So between two images where point A (50m altitude) moved 600m, and point B (0m altitude) has moved
> 0.005 * 20 = 0.1 m.
> Hence, the parallax effect is imperceptible for 50m tall objects. Even less so for the objects that we have underlined above, and our results support this.
>
> There is indeed a limitation for objects (e.g. mountains, and towering clouds) whose depth variation (well beyond 50m) indeed leads to significant parallax effects. Thank you for pointing out this limitation of MFSR for remote sensing. We will include it in our discussion.

---

> ### Author Response · Authors · 2019-11-12
> **Discussion of various concerns and clarifications**
>
> Below we address various concerns and provide some clarifications
>
> 1. > "I am not convinced that pair-wise fusion can handle significant translational fusion as the filters have shared parameters"
>
> Indeed such an approach would be problematic. However, this is not what we proposed. We have already addressed this concern in a previous comment. Please see paragraph 4 of this comment:
> https://openreview.net/forum?id=HJxJ2h4tPr&noteId=B1l6sKmUoB
> "Please note that the recursive fusion stage accepts only the encoded low-res / reference pairs. So nothing in our co-registration scheme explicitly assumes that the difference in low-res images must be explained by translational motion only."
>
> ---
>
> 2. > "How a single convolutional layer accomplishes a global encoding"
>
> Our fusion block consists of 3 convolutional layers, not a single one. See section 3.1, paragraph "Fuse", and Table 3 in Appendix A1.
>
> ---
>
> 3. > "Of course, such a problematic approach needs at least some kind of motion compensation"
>
> We hope our comments helped clarify our approach, and we thank the reviewer for bringing up these concerns.
>
> ---
>
> 4. > "Even assuming the method only applies to satellite imagery"
>
> We focus on PROBA-V for a few reasons:
> - Prior SR work focuses on super-resolving low-res images that are artificially generated by simple bilinear down-sampling, see "To learn image super-resolution, use a GAN to learn how to do image degradation first", Bulat, et al. ECCV18.
> We forgot to mention (and we'll include this) that PROBA-V has separate cameras onboard for capturing high-res / low-res pairs. We are not aware of other datasets with real low-res images (not synthetically down-scaled). As far as we know, the PROBA-V dataset is the first publicly available dataset for MFSR that contains naturally occurring low-res and high-res pairs . This is in contrast to most of the work in SR (SISR, MFSR, Video SR, Stereo SR) that synthetically down-sample high-res images. See also, image restoration track at CVPR19, where the vast majority of challenges are performed on synthetically downscaled / degraded images: REDS4:  "Ntire 2019 challenge on video deblurring: Methods and results", NTIRE Workshop at CVPR 2019, http://www.vision.ee.ethz.ch/ntire19
> - Methods that are trained on artificially downscaled datasets fail to produce good results when applied to real-world low-resolution, low quality images (Bulat, et al. ECCV18; Zero-shot SR using Deep Internal Learning, CVPR18). For this reason we experimented only on PROBA-V, a dataset that does not suffer from biases induced by artificial down-sampling.
>
> ---
>
> 5. > "Characterization of satellite imagery noise models (Weibull, etc.) common in such imagery as a prior also completely disregarded."
>
> Results from DeepSUM and our HighRes-net (top methods in the ESA competition) and ESA's MISR paper (“Super-resolution of PROBA-V images using CNNs.” Astrodynamics, 2019) - all suggest that neural networks can learn the data-driven representations for MFSR, without the need to characterize noise models explicitly. For one, the form of the loss function itself implicitly assumes a noise model: a Gaussian for an MSE / L2-norm loss, a Laplace for a L1-norm loss. That said, we agree with you that using an explicit noise model in the architecture can be valuable depending on the application, e.g. Weibull for SAR satellite imagery ("Modeling SAR images with a generalization of the Rayleigh distribution", IEEE TIP 2004; "Learning to detect roads in high-resolution aerial images", ECCV 2010)
>
> ---
>
> 6. > "it lacks mechanisms to compensate/distinguish cloud coverage"
>
> Clouds and other volatile objects (e.g. snow) can be accounted for with cloud masks - binary masks that indicate missing / occluded values in an image. Such masks can be added as an input channel, for every low-res view, as we did for the reference frame.
> When missing value masks are available, neural networks can learn which part of the input are anomalous, noisy, or missing from such binary masks. See e.g. "Recurrent neural networks for multivariate time series with missing values", Che et al., Scientific reports 8.1 (2018)
>
> The ESA PROBA-V dataset provides clouds masks for every low-res view. We use them to bias our model towards sampling low-res views that are likely to have high "clearance", see appendix A.4.
> One omission, is that we did not define "clearance: Clearance is the fraction of occluded (clouded) values in an image, as per the cloud mask.
> We will include this in our final version.
>
> In satellite applications where clouds masks are not available, other segmentation methods are in order to infer such masks, e.g. "U-net, Ronneberger, et al., MICCAI, 2015.
> More generally, cloud detection can be considered a preliminary preprocessing step.
>
> We have amended the discussion in Section 5.1
>
> ---
>
> 7. > "atmospheric distortions"
>
> Please see paragraph 5 in this comment
> https://openreview.net/forum?id=HJxJ2h4tPr&noteId=B1l6sKmUoB

---

> ### Author Response · Authors · 2019-11-13
> **Minor clarifications**
>
> Thank you for bringing points that were unclear to our attention.
>
> 1. > "It first estimates a reference image for the multiple input LR images by median filtering."
>
> Our reference image is not computed by median filtering. A median filter replaces each pixel with the median of its neighborhood (see e.g. Marion, "An Introduction to Image Processing", p 274.)
>
> Our reference frame is actually defined as the median across the LR set, please see section 3.1, equation 1.
> As Reviewer 1 suggests
> reference_image($i, j$) $=$ median $( LR1(i, j), LR2(i, j), \cdots)$
>
> Thank you for raising this.
> We have clarified this in equation 1, section 3.1.
>
> ---
>
> 2. > "Then it pairwise encodes the reference image and each of the multiple images in a recursive fashion"
>
> The initial encoding itself is not recursive. Each reference-LR pair is encoded individually, see equation 2. The recursion happens later, during the fusion of all encodings, please see equations 3 and 4 and figure 4.

---

### Decision · Program_Chairs · 2019-12-19

**Decision:**

Reject

**Comment:**

This paper proposes a multi-frame super-resolution method including recursive fusion for co-registration and registration loss to solve the problem where the super-resolution results and the high-resolution labels are not pixel-wise aligned. While reviewer #1 is positive about this paper, reviewer #2 and #3 rated weak reject and reject respectively. Both reviewer #2 and #3 have extensive experience in the topic of image super-resolution. The major concerns raised by the reviewers include the lack of many references, the comparison of recursive fusion with related work, limited test databases, using a single translational motion for the SR images, and limited novelty on the network modules.  The authors provided detailed response to the concerns, however they did not change the overall rating of the reviewers. While the ACs agree that this work has merits, given the various concerns raised by the reviewers, this paper can not be accepted at its current state.

---

> ### Author Response · Authors · 2019-12-22
> **Lack of due diligence in meta-review. Simply re-asserting criticisms, ended by a generic canned rejection response.**
>
> Disclaimer:
> I have emailed the PCs. I am now publicly posting the same by recommendation of Kyunghyun Cho (PC). Their recommendation does not imply endorsement of what I'm about to say.
>
> --------------
>
> Dear Program Chairs,
>
> I appreciate that the workload of the ACs has been intense, given the increased number of submissions this year. I was surprised to hear that even PCs had to take on some meta-reviews. Regardless, the AC should encourage the reviewers to engage in discussion, and maintain a constructive standard in meta-reviews. In the face of disagreement (scores 1,3,8), the ACs should have assigned a 4th reviewer.
>
> We find the AC decision extremely unfair, given that the reviewers were not responsive, and the meta-review simply re-asserts the criticisms without acknowledging our rebuttal. This approach is unscientific.
>
> To make matters worse, the meta-review closed with a canned generic response:
>
> "The authors provided detailed response to the concerns, however they did not change the overall rating of the reviewers. While the ACs agree that this work has merits, given the various concerns raised by the reviewers, this paper can not be accepted at its current state."
>
> This could apply to *any* rejection, and shows lack of due diligence.
>
> We addressed every single point of criticism and misconception in our rebuttal. But the reviewers were not responsive, with the exception of a clarification from R3.
>
> • we added missing references suggested by R2, and more.
>
> • R3 was not aware of the "parallax" term - a concept so basic in Remote Sensing and Super-Resolution. They might be an expert, but certainly not in the immediate area of this paper. The ACs did not respond when we flagged this. That said, R3's concerns remain valid, and in fact lead to a useful discussion, that was unfortunately ignored by ACs.
>
> • R3 made a short and vacuous critisism on novelty without explaining in detail their reasoning nor listing related literature. The meta-review not only did not acknowledge this, but also re-asserted the same unqualified statement.
>
> • the comparisons/datasets that were suggested are not applicable because they address different problem settings. We explained why that is - yet again no response from but reviewers & ACs.
>
> • the co-registration of low-res images *is not* translational. This was a misconception. We have commented on this extensively. The final SR image is translated only during training to get a more accurate loss estimate. The translation is a nuisance parameter. Again, our comments were ignored.
>
> • Our paper is the state of the art in the European Space Agency competition, which uses a real-world satellite dataset, not a toy-dataset made of artificial low-res images. The ESA PROBA-V dataset is the only MFSR dataset that contains naturally occurring high-res and low-res imagery.
>
> The ACs did not acknowledge any of this. From the meta-review, it is not clear if they've read our rebuttal. The fact that the reviewers did not increase their scores, does not mean that we did not address these critisisms.
>
> An AC should never blindly trust the self-assessed expertise of the reviewers, and they should call out incomplete reviews. An AC should read the reviews and the rebuttal, and should be able to assess which concerns have been addressed.
>
> Sincerely,
> Alfredo